# PALATE: Peculiar Application of the Law of Total Expectation to Enhance the Evaluation of Deep Generative Models

## Abstract

Deep generative models (DGMs) have caused a paradigm shift in the field of machine learning, yielding noteworthy advancements in domains such as image synthesis, natural language processing, and other related areas. However, a comprehensive evaluation of these models that accounts for the trichotomy between fidelity, diversity, and novelty in generated samples remains a formidable challenge. A recently introduced solution that has emerged as a promising approach in this regard is the Feature Likelihood Divergence (FLD), a method that offers a theoretically motivated practical tool, yet also exhibits some computational challenges. In this paper, we propose PALATE, a novel enhancement to the evaluation of DGMs that addresses the limitations of FLD regarding computational efficiency. Our approach is based on a peculiar application of the law of total expectation to random variables representing accessible real data. When combined with the MMD baseline metric and DINOv2 feature extractor, PALATE offers a holistic evaluation framework that matches or surpasses state-of-the-art solutions while providing superior computational efficiency and scalability to large-scale datasets. Through a series of experiments, we demonstrate the effectiveness of the PALATE enhancement, contributing a computationally efficient, holistic evaluation approach that advances the field of DGMs assessment, especially in detecting sample memorization and evaluating generalization capabilities.

## 1 Introduction

In recent years, deep generative models (DGMs) have garnered significant attention, becoming a subject of interest not only for the machine learning community, but also for researchers across other scientific disciplines, practitioners, and even the general public (Ravuri et al., 2023). DGMs have already reached a sufficient level of maturity for utilization in downstream tasks, leading to the generation of substantial results. These include, but are not limited to, photorealistic imagery (Rombach et al., 2022), verbal expressions that emulate authentic discourse (Goel et al., 2022), and written text reminiscent of human composition (OpenAI, 2023).

Regardless of the architectural framework employed, a critical component of any deep generative model is the generator, i.e., a network meticulously designed and trained to produce synthetic data that are indistinguishable from real data. Although the learning of DGMs typically entails an optimization process for a model-specific objective function, a fair post-learning assessment necessitates the utilization of an adequate approach that is not biased towards any particular underlying architecture. This is usually achieved by implementing an appropriate technique that utilizes samples of generated data, otherwise referred to as "fake data," preferably in relation to the real data. Consequently, in such a case, it can be asserted that we are dealing with a sample-based evaluation metric. The most prominent state-of-the-art examples of such metrics include inception score (IS) (Salimans et al., 2016) and Fréchet inception distance (FID) (Heusel et al., 2017). The popularity of IS and FID stems from their quite reasonable consistency with human perceptual similarity judgment, ability to recognize diversity within generated samples, and ease of use. Additionally, FID has been shown to effectively capture sample fidelity, a consequence of its definition as a statistical distance between feature distributions appropriately fitted to real and generated data samples. Nevertheless, despite its treatment as a gold evaluation standard, FID generally lacks the capacity to detect overfitting

or sample memorization (Jiralerspong et al., 2024). This limitation has not been resolved by more recent, somewhat popular approaches such as kernel inception distance (KID) (Bińkowski et al., 2018) or precision and recall (P&R) (Sajjadi et al., 2018), which were otherwise designed to add nuance to the evaluation process by addressing some other weaknesses of FID, such as strong bias, normality assumption, or the inability to capture fidelity and diversity as separate characteristics. Instead, the extant proposals have encompassed the implementation of autonomous metrics, such as the $C_T$ score (Meehan et al., 2020) and authenticity (Alaa et al., 2022). The development of these approaches has been undertaken with the objective of detecting overfitting (or sample memorization) and consequently evaluating the generalization properties of DGMs.

While the significance of a thorough examination of the quality of the outcomes of generative models cannot be overstated, this subject appears to be underrepresented and underestimated in the literature. Among the various properties of a good evaluation metric, the most important seems to be its holistic nature, which can be understood as the ability to validate generated samples in a trichotomous way involving their fidelity, diversity, and novelty—a newly defined concept opposite to memorization. The most recent and promising advancement in this field was proposed by the authors of (Jiralerspong et al., 2024), who introduced the Feature Likelihood Divergence (FLD), especially designed to verify generalization capabilities while preserving the advantages of existing approaches. Nevertheless, while FLD has proven to be a viable metric for holistic evaluation, it has been observed to encounter computational challenges when applied to more complex and varied real-world datasets. This is due to the fact that for such datasets, it is necessary to use a substantial number of data samples to calculate reliable metric values, which appears to be computationally demanding (see the results from our experiments on the large-scale ImageNet dataset (Deng et al., 2009) presented in Section 5). In order to address this issue, we propose a novel enhancement to the evaluation of deep generative models, which we refer to as PALATE. Our solution is grounded in a **p**eculiar **a**pplication of the **la**w of **t**otal **e**xpectation (hence the name). The objective of the PALATE enhancement is to deliver a technique that improves upon a given baseline metric capable of validating the fidelity and diversity of generated samples, to account for their novelty. Our experimental results demonstrate that this approach, when implemented in conjunction with a DINOv2 version of the recently developed CLIP-MMD (CMMD) metric (Jayasumana et al., 2024), which we call DINOv2-MMD (DMMD), facilitates the creation of a holistic evaluation metric that is aligned with state-of-the-art solutions and exhibits superior computational efficiency.

In summary, our work contributes the following: *(i) we propose PALATE, a novel enhancement to the evaluation of DGMs*, which is designed to improve a baseline metric to ensure its sensitivity to the memorization of training samples, *(ii) we present a theoretical justification for our approach*, which derives from the law of total expectation applied in a peculiar way to random variables representing all available real data, *(iii) we conduct extensive experiments on real-world data*, which confirm the usefulness of our approach to provide a computationally efficient holistic evaluation metric that matches (or even surpasses) state-of-the-art solutions.

## 2 PRELIMINARIES

This section summarizes the basic concepts underlying generative models and explains the law of total expectation, an essential tool for the approach we propose.

**Generative Models** In this work, we consider a scenario in which we have access to two distinct collections of real data: a train dataset and a test dataset, with the former dedicated for training and the latter for evaluation purposes exclusively. These data can be considered independently drawn from a random variable $X$ acting on a given multidimensional Euclidean data space $\mathcal{X}$. Additionally, following (Bengio et al., 2013; Loaiza-Ganem et al., 2024), it is assumed that all accessible data lie on a lower dimensional manifold $\mathcal{M}_{\text{data}} \subset \mathcal{X}$, i.e., $\mathbb{P}(X \in \mathcal{M}_{\text{data}}) = 1$.

Despite the existence of numerous generative architectures, each of them contains an indispensable part that plays the role of a generator—henceforth denoted by $G$—designed and trained to produce synthetic data that are indistinguishable from real data. These generated data can also be regarded as being drawn independently from a random variable $Y$ defined on the common data space $\mathcal{X}$, which follows another distribution $p_G$ that is determined by the model. It is important to note that while the real data distribution $p_X$, though unknown, remains fixed, the fake data distribution $p_G$ is subject to change during the training process. The objective is to force $p_G$ to approximate $p_X$ as closely

as possible, a goal that is typically achieved by optimizing a model-specific loss function utilizing training data samples. On the other hand, when attempting to draw fair comparisons among various optimized generative models, a model-agnostic evaluation metric is required, one that can effectively access the learning effects. Such a metric can be constructed to consider either only the properties of $p_G$ (as IS) or the discrepancy between $p_X$ and $p_G$ (as FID). This is typically accomplished by leveraging respective test and generated data samples, resulting in a sample-based metric. In addition, the prevailing approaches employ an ancillary feature extractor $\phi$, such as the pre-trained Inception-v3 (Szegedy et al., 2016), DINOv2 (Oquab et al., 2023), or CLIP (Radford et al., 2021) network. The objective of this network is to embed samples into a perceptually meaningful feature space, thereby reducing the task to the assessment of the feature distributions $p_{\phi(X)}$ and $p_{\phi(G)}$. In the subsequent section, we present our selection of the most salient state-of-the-art evaluation metrics for DGMs, which are pertinent to our work.

**Law of Total Expectation**   The law of total expectation (Kolmogorov, 1933; Feller, 1968), alternatively referred to as the law of iterated expectations or the tower rule, is a foundational principle in probability theory. Its application to a partition of the sample space proves particularly advantageous, as it facilitates computation of the expected value of a random variable through its decomposition into conditional expectations over mutually exclusive and exhaustive events. This is the subject of the following theorem.

**Theorem 1** *Let $Z$ be a random variable with finite expectation, and let $\{A_1, \ldots, A_n\}$ be a partition of a sample space $\Omega$,* i.e., $\bigcup_{i=1}^n A_i = \Omega$ and $A_i \cap A_j = \varnothing$ for $i \neq j$, with $\mathbb{P}(A_i) > 0$ for all $i$. Then *the following equality holds:*

$$\mathbb{E}(Z) = \sum_{i=1}^n \mathbb{E}(Z|A_i)\mathbb{P}(A_i). \tag{1}$$

For the convenience of the reader, a simplified proof of Theorem 1 is provided in Appendix C.

## 3   RELATED WORK

In this section, we present state-of-the-art evaluation metrics for DGMs that are relevant to our work.

**Fidelity and Diversity Metrics**   Inception score (IS) (Salimans et al., 2016) is a sample-based metric, which utilizes the pre-trained Inception-v3 network (Szegedy et al., 2016) to evaluate the diversity of generated samples. However, IS is constrained in its ability to assess the fidelity of these samples to real data. To address this limitation, several metrics have been proposed that focus on both fidelity and diversity. These include Fréchet inception distance (FID) (Heusel et al., 2017), kernel inception distance (KID) (Bińkowski et al., 2018), and CLIP-MMD (CMMD) (Jayasumana et al., 2024) metrics. FID utilizes the feature distributions of real and generated data, extracted using the Inception-v3 network, to calculate the Wasserstein distance between them, thereby offering a means to evaluate both fidelity and diversity. As part of a broader evaluation framework, in (Stein et al., 2024) the authors introduce $FD_{DINOv2}$, a variant of FID that uses DINOv2 features, which better capture global structure and salient objects, improving perceptual alignment. While FID is widely accepted as the gold evaluation standard, it is sensitive to the biases inherent in the inception space and suffers from approximation of features of both real and generated data by a normal distribution. Conversely, KID is a metric based on maximum mean discrepancy (MMD) between the feature distributions, which does not assume normality and is less affected by biases. CMMD, another recently proposed metric, also utilizes MMD but employs the Gaussian RBF characteristic kernel and CLIP embeddings instead of the rational quadratic kernel and Inception-v3 embeddings used by KID. Improved Precision and Recall (Kynkäänniemi et al., 2019) further decompose sample quality and coverage using non-parametric manifolds, offering more nuanced evaluation than FID. Density and Coverage (Naeem et al., 2020) offer an empirically reliable and theoretically grounded approach that successfully addresses the limitations of Precision and Recall. However, these metrics are ineffective in addressing the challenges posed by memorization and overfitting in DGMs. Other recently proposed evaluation metrics that more reliably quantify fidelity and diversity include Vendi Score (Friedman & Dieng, 2023), FKEA (Ospanov et al., 2024), and RKE Score (Jalali et al., 2023). The Vendi Score employs a diversity measure grounded in kernel methods, offering robust diversity assessment. FKEA introduces a scalable, reference-free evaluation leveraging feature kernel approximations for fidelity and diversity, while the RKE Score uses information-theoretic criteria to

evaluate multi-modal distributions. These recent advances represent significant progress in capturing generative model performance beyond classical metrics.

**Holistic Metrics** The field of holistic evaluation of DGMs has recently emerged as a novel approach to assessing their generalization capabilities, which has been achieved by simultaneously considering multiple aspects of sample quality. A notable example is Feature Likelihood Divergence (FLD) (Jiralerspong et al., 2024), which integrates fidelity, diversity, and novelty into a single metric. FLD employs a kernel density estimator (KDE) with an isotropic Gaussian kernel to model the feature distribution of generated data. Subsequently, KDE is applied to test data features extracted using networks such as Inception-v3 or DINOv2. It is noteworthy that the KDE bandwidth matrix is optimized to penalize the memorization of training samples, thereby enabling FLD to detect overfitting while maintaining other benefits. However, FLD faces challenges with complex real-world datasets, as it requires a large number of samples to produce reliable results, which can be computationally demanding.

**Memorization Metrics** Concerns regarding the tendency of DGMs to memorize training data and overfit have given rise to the development of metrics capable of detecting such behaviors. Notable examples include $C_T$ score (Meehan et al., 2020) and authenticity (Alaa et al., 2022). These metrics are designed to identify instances where generated samples exhibit a high degree of similarity to training data, suggesting a potential for memorization. Specifically, the $C_T$ score quantifies the probability that a generated sample will be indistinguishable from a real sample, thereby helping to detect overfitting. On the other hand, authenticity is determined through a binary sample-wise test to ascertain whether a sample is authentic (i.e., overfit), and is typically expressed as the AuthPct score, which quantifies the percentage of generated samples classified as authentic. In this context, it is also pertinent to mention Generalization Gap FLD, which is a memorization metric calculated by subtracting the value of FLD from its version computed using the train dataset instead of the test dataset. Novelty assessment has further benefited from metrics like FINC (Zhang et al., 2025), which employs scalable differential clustering to detect generation uniqueness. On the other hand, complementary metrics such as the Rarity Score (Han et al., 2022) and KEN score (Zhang et al., 2024) aim to quantify the uniqueness and informational entropy of generated outputs in relation to the training set. While these metrics provide important insights into memorization and overfitting, they do not fully capture the holistic quality of generated samples that includes fidelity and diversity. This motivates the design of evaluation frameworks that consider all these aspects jointly.

## 4 PALATE ENHANCEMENT

This section presents a novel enhancement for the evaluation of DGMs, referred to as PALATE, which leverages a **p**eculiar **a**pplication of the **la**w of **t**otal **e**xpectation (hence the name). The objective is to enhance existing baseline methods by accounting for the memorization of training samples, thereby creating a holistic evaluation metric consistent with state-of-the-art solutions such as FLD. The following paragraphs outline the general formulation of our approach and discuss relevant hyperparameters.

**PALATE** The following assumptions underpin our approach: *(A1)* for the samples of training data $x_{\text{train}} = \{x_1^{\text{train}}, \ldots, x_m^{\text{train}}\}$ and test data $x_{\text{test}} = \{x_1^{\text{test}}, \ldots, x_n^{\text{test}}\}$, which are selected for evaluation, we have two non-trivial disjoint parts[1] $\mathcal{M}_{\text{train}}$ and $\mathcal{M}_{\text{test}}$ of the manifold of data $\mathcal{M}_{\text{data}}$, such that $\mathcal{M}_{\text{data}} = \mathcal{M}_{\text{train}} \cup \mathcal{M}_{\text{test}}$, $x_{\text{train}} \subset \mathcal{M}_{\text{train}}$, and $x_{\text{test}} \subset \mathcal{M}_{\text{test}}$, *(A2)* the ratio of the cardinalities of $x_{\text{train}}$ and $x_{\text{test}}$ (i.e., $m/n$) remains constant across all models and datasets, *(A3)* a baseline metric $M_{\text{base}}$ capable of capturing the fidelity and diversity of generated samples is defined by the conditional expectation operator, i.e., $M_{\text{base}} = \mathbb{E}(Z|X \in \mathcal{M}_{\text{test}})$ for some random variable $Z$ with $\mathbb{E}(Z) = 0$ if and only if $p_X = p_G$ (note that any such $Z$ must implicitly depend on the random variables $X$ and $Y$—see Equation (5) for an example). Given these conditions, the law of total expectation, as stated in Theorem 1, can be applied to $Z$ and the partition $\Omega = \{\omega \mid X(\omega) \in \mathcal{M}_{\text{test}}\} \cup \{\omega \mid X(\omega) \in \mathcal{M}_{\text{train}}\}$, yielding the following formula:

$$\mathbb{E}(Z) = \mathbb{E}(Z|X \in \mathcal{M}_{\text{data}}) = a\,\mathbb{E}(Z|X \in \mathcal{M}_{\text{test}}) + (1 - a)\,\mathbb{E}(Z|X \in \mathcal{M}_{\text{train}}), \tag{2}$$

---

[1]We emphasize that these parts do not require any special structure (specifically, they do not need to be submanifolds or connected sets), are induced solely by the data samples selected for evaluation, and are not predetermined or fixed subsets.

where $a = \mathbb{P}(X \in \mathcal{M}_{\text{test}}) = 1 - \mathbb{P}(X \in \mathcal{M}_{\text{train}})$. It is crucial to acknowledge that Equation (2) integrates the baseline approach (the first right-hand side term), which is predicated on fidelity and diversity, with the concept of novelty (the second right-hand side term) in a concise formula. Specifically, this expression enables the recognition of the extent to which the generation of samples by an optimized model that closely align with the manifold of data is attributable to overfitting or even replication of training data samples. The assessment of this phenomenon can be facilitated by employing the following formula:

$$\text{PALATE}(M_{\text{base}}) = \frac{\mathbb{P}(X \in \mathcal{M}_{\text{test}})\, \mathbb{E}(Z|X \in \mathcal{M}_{\text{test}})}{\mathbb{E}(Z|X \in \mathcal{M}_{\text{data}})} = \frac{a\, \mathbb{E}(Z|X \in \mathcal{M}_{\text{test}})}{a\, \mathbb{E}(Z|X \in \mathcal{M}_{\text{test}}) + (1-a)\, \mathbb{E}(Z|X \in \mathcal{M}_{\text{train}})}. \tag{3}$$

This definition clearly shows that the value yielded by Equation (3) approaches 1 (i.e., the maximum) for the copycat model, which merely samples from the train dataset, while for an optimized model— that is, one that attains a superior baseline metric score—it has been minimized. Consequently, we can conclude that DGMs exhibiting a tendency to memorization (or overfitting) are those for which $\text{PALATE}(M_{\text{base}}) > a$ (see Appendix C, where we also provide complementary theoretical study relating the PALATE approach to the classical concept of data-copying proposed in (Meehan et al., 2020)).

**PALATE Enhancement**  As discussed above, there is a compelling argument for adopting $\text{PALATE}(M_{\text{base}})$ as an alternative evaluation metric, as it demonstrates significant advances in terms of recognizing memorization of training data samples when compared to $M_{\text{base}}$. Nevertheless, it is crucial to acknowledge the ambiguity that it introduces when differentiating between well-optimized and poorly-optimized models. Specifically, it is a typical occurrence for $\mathbb{E}(Z|X \in \mathcal{M}_{\text{test}})$ to approximate $\mathbb{E}(Z|X \in \mathcal{M}_{\text{train}})$, irrespective of the quality of the model, which leads to score flattening[2]. To address this issue, we propose to strengthen the impact that $M_{\text{base}}$ has on the final metric value. The PALATE enhancement can therefore be delineated in terms of a weighted average of the baseline metric score (scaled to $[0, 1]$) and the value provided by Equation (3), i.e.:

$$M_{\text{PALATE}}(M_{\text{base}}) = \alpha\, \text{SCALE}(M_{\text{base}}) + (1 - \alpha)\, \text{PALATE}(M_{\text{base}}), \tag{4}$$

where $\alpha \in [0, 1)$ is a weighting constant. Note that such a general formula relies on various hyperparameters, the selection of which is discussed in the following paragraphs.

**Baseline Metric and Scaling Method**  Since the baseline metric is assumed to be defined as the expectation of a random variable, the most suitable candidates are those based on the maximum mean discrepancy (MMD) (Gretton et al., 2006) between two probability distributions $p$ and $q$, which can be derived from the following general formula:

$$\text{MMD}_k^2(p, q) = \mathbb{E}(Z) \ \text{for} \ Z = k(X_1, X_2) + k(Y_1, Y_2) - 2\, k(X_1, Y_1), \tag{5}$$

where $X_1, X_2$ and $Y_1, Y_2$ are independently distributed by $p$ and $q$, respectively, and $k(\cdot, \cdot)$ is a given positive definite kernel. In the context of evaluation of DGMs, Equation (5) is employed for feature distributions $p_{\phi(X)}$ and $p_{\phi(G)}$. State-of-the-art examples include KID and (more recent) CMMD metrics, which were delineated in Section 3. We opt to use CMMD as a baseline metric due to its superiority over KID, as it utilizes the characteristic Gaussian RBF kernel $k^{\text{RBF}}(x, y) = \exp(-\frac{1}{2\sigma^2}\|x - y\|^2)$, which renders MMD a distribution-free statistical distance and facilitates the following straightforward scaling technique:

$$\text{SCALE}(\text{MMD}_{k^{\text{RBF}}}^2(p, q)) = \frac{\text{MMD}_{k^{\text{RBF}}}^2(p, q)}{\mathbb{E}(k^{\text{RBF}}(X_1, X_2)) + \mathbb{E}(k^{\text{RBF}}(Y_1, Y_2))} \in [0, 1]. \tag{6}$$

Furthermore, a specific value of the bandwidth parameter proposed in (Jayasumana et al., 2024), i.e., $\sigma = 10$, is maintained. However, in our approach, we abandon the use of CLIP embedding network, as in the case of CMMD, but employ DINOv2 as the feature extractor. Thus, the baseline metric is designated as DINOv2-MMD (DMMD), rather than CMMD. This substitution is motivated by the fact that CLIP lacks the fine-grained recognition capabilities, which is due to the poor separability of object characteristics in the CLIP latent space (Bianchi et al., 2024). It is important to note that the ability to distinguish between subtle object features like color and shape seems to be crucial when we try to detect overfitting or memorization. On the other hand, DINOv2 feature space has been demonstrated to facilitate a more comprehensive evaluation of DGMs and to exhibit stronger correlation with human judgment in comparison to Inception-v3, as asserted by the authors of (Stein et al., 2024), which further supports our choice.

---

[2]This is due to the fact that both training and test data are assumed to follow the same data distribution $p_X$.

Table 1: Evaluation of different DGMs on CIFAR-10 and ImageNet. Our proposed metrics are shown in blue. The remaining results are rewritten from (Jiralerspong et al., 2024; Stein et al., 2024).

| Dataset | Model | Human error rate ↑ | $M_{\text{PALATE}}$ ↓ | FLD ↓ | $FD_{\text{DINOv2}}$ ↓ | PALATE ↓ | Gen. Gap $FD_{\text{DINOv2}}$ ↑ | $C_T$ ↑ | AuthPct ↑ |
|---|---|---|---|---|---|---|---|---|---|
| | | | Holistic metrics | | | Memorization metrics | | | |
| CIFAR-10 | PFGM++ | 0.436 ± 0.011 | 0.7079 | 4.58 | 80.47 | 0.4999 | -0.59 | 32.79 | 83.54 |
| | iDDPM-DDIM | 0.400 ± 0.013 | 0.7140 | 5.63 | 128.57 | 0.5003 | -0.56 | 39.65 | 84.60 |
| | StyleGAN-XL | 0.399 ± 0.012 | 0.7217 | 5.58 | 109.42 | 0.4995 | -0.37 | 36.79 | 85.29 |
| | StyleGAN2-ada | 0.393 ± 0.012 | 0.7265 | 6.86 | 178.64 | 0.4997 | -0.24 | 45.31 | 86.40 |
| | BigGAN-Deep | 0.387 ± 0.014 | 0.7282 | 9.28 | 203.90 | 0.4995 | -0.06 | 55.70 | 88.10 |
| | MHGAN | 0.336 ± 0.015 | 0.7342 | 8.84 | 231.38 | 0.4997 | -0.19 | 47.87 | 86.69 |
| | LOGAN | 0.206 ± 0.020 | 0.7486 | 6.07 | 881.73 | 0.5070 | 0.18 | 55.66 | 84.10 |
| | ACGAN-Mod | 0.148 ± 0.013 | 0.7486 | 24.22 | 1143.07 | 0.4994 | 0.17 | 26.11 | 72.09 |
| ImageNet 256×256 | LDM | 0.309 ± 0.017 | 0.6061 | 3.41 | 82.42 | 0.5042 | -0.74 | 33.63 | 69.23 |
| | DiT-XL-2 | 0.286 ± 0.016 | 0.5919 | 1.98 | 62.42 | 0.5058 | -0.99 | 22.57 | 65.79 |
| | RQ-Transformer | 0.223 ± 0.012 | 0.7174 | 11.55 | 212.99 | 0.5016 | -0.53 | 125.48 | 86.10 |
| | Mask-GIT | 0.183 ± 0.016 | 0.6923 | 6.74 | 144.23 | 0.5025 | -0.63 | 78.97 | 80.02 |
| | GigaGAN | 0.16 ± 0.01 | 0.7390 | 8.34 | 156.40 | 0.5001 | -0.42 | 98.78 | 82.48 |
| | StyleGAN-XL | 0.153 ± 0.013 | 0.7050 | 8.46 | 150.27 | 0.5017 | -0.40 | 98.69 | 84.10 |

**Splitting of Datasets**    Despite the proposed approach being predicated on the division of the data manifold into two disjoint parts related to the selected samples of training and test data, in this case, only the value of $a = \mathbb{P}(X \in \mathcal{M}_{\text{test}}) \in (0, 1)$ is required to perform all of the computations. Given that the data distribution $p_X$ is unknown, the sole method is to treat $a$ as a hyperparameter and estimate it from the given data samples. This can be achieved by applying the law of total expectation once again, but this time to a one-dimensional embedding of the random variable $X$. The value of $a$ should thus be established as a fraction of test data samples within all data samples selected for evaluation, i.e., $a = n/(m + n)$ (note that, due to assumption (A2), the value of $a$ remains constant across all models and dataset). A detailed proof of this formula can be found in Appendix C. The other factors influencing computations are sample sizes $m$, $n$, and $k$, utilized for the train, test, and generated datasets, respectively. We recommend using samples of equal size ($m = n = k$), as this is a common approach for evaluating DGMs—note that this choice fixes $a = 1/2$. However, the PALATE enhancement, in contrast to most state-of-the-art solutions, also incorporates training data samples, which is in pair with the FLD metric. It is imperative to note that, for the calculation of FLD, it is crucial to utilize sufficient number training samples, as otherwise it is possible to obtain negative metric values. This is due to the necessity of employing a portion of the train dataset to estimate a particular dataset-dependent constant, as outlined in (Jiralerspong et al., 2024). Consequently, such an issue influences the computational efficiency of FLD, as confirmed by our experiments presented in Section 5.

**Weighting Constant**    The weighting constant $\alpha$ signifies the degree to which our metric prioritizes memorization, thus affecting the sensitivity to fidelity and diversity in the generated samples. While its value can be set arbitrarily, depending on the specific goal, we examine two notable cases: $\alpha = 0$ and $\alpha = 1/2$. In the first case, the metric is maximally oriented towards the detection of sample memorization, while in the second case it balances this capacity equally with other abilities. Therefore, our work focuses on these specific values of the hyperparameter $\alpha$.

In the last paragraph of this section, we address the issue of estimating the proposed evaluation metrics. Then, the experimental analysis is provided in Section 5.

**Estimation of Metrics**    As noted above, our primary focus is on two evaluation metrics, namely PALATE (memorization metric) and $M_{\text{PALATE}}$ (holistic metric), derived from Equations (3) and (4) with $\alpha = 0$ and $\alpha = \frac{1}{2}$, respectively. We use DMMD as the baseline metric, with the SCALE function given in Equation (6). The next step is to compute metric values based on selected real and generated data samples $x^{\text{train}}$, $x^{\text{test}}$, and y. To this end, we propose to replace all kernel expectations with their respective V-statistics, a common practice in MMD estimation (see, e.g., (Gretton et al., 2012)). In Appendix D, we provide direct formulas for the PALATE and $M_{\text{PALATE}}$ metrics, along with implementation details.

At the end of this section, we discuss the relationship of our approach with traditional data splitting and cross-validation concepts.

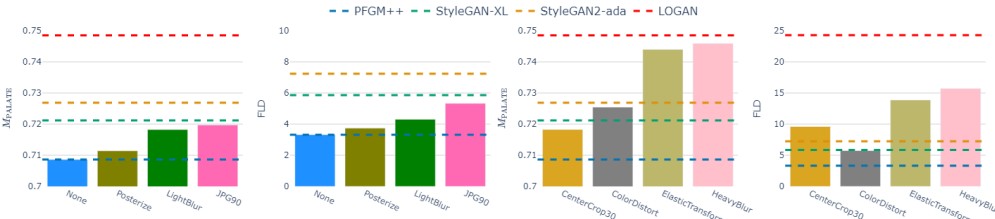

Figure 1: Comparison of the effects of different transformations, applied to samples generated by PFGM++ trained on CIFAR-10, on $M_{\text{PALATE}}$ and FLD (corresponding values for other models are provided for reference). *Left*: Nearly imperceptible transformations. *Right*: Large transformations.

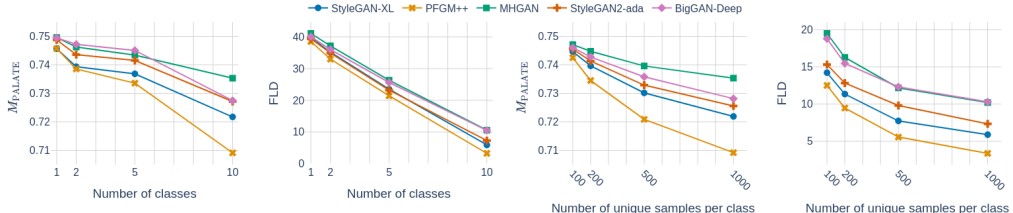

Figure 2: Capability of $M_{\text{PALATE}}$ (our) and FLD to capture sample diversity in two experimental settings. *Left*: Varying the number of classes while maintaining a fixed total sample size of 10000 by adjusting the duplication of 1000 fixed samples per class. *Right*: Varying the number of unique samples per class, with equal replication across classes to maintain class balance and a total sample size of 10000.

**Relationship with data splitting and cross-validation.** The PALATE approach does not introduce the concept of data splitting per se but rather incorporates the train-test split within a comprehensive evaluation framework, similar to FLD, which, to the best of our knowledge, was the first metric to employ this approach for jointly assessing fidelity, diversity, and memorization. Unlike cross-validation, which is a training technique used to estimate model generalization during learning (Hastie et al., 2009), our method uses a train-test split solely for evaluation purposes of deep generative models. Traditional evaluation metrics often treat the entire dataset as a whole without explicitly distinguishing between training and testing subsets, thereby limiting assessment to fidelity and diversity while omitting memorization and overfitting analysis. By explicitly integrating the split into the evaluation metric, the resulting holistic metric $M_{\text{PALATE}}$ quantifies how closely generated samples resemble training data versus novel test data, addressing a critical gap in classical frameworks. In summary, the PALATE approach builds on the concept of data splitting but does not claim novelty in splitting itself. Instead, it applies the split within a principled, post-learning evaluation metric that captures memorization, an aspect not directly addressed by classical whole-dataset evaluations.

## 5 EXPERIMENTS

In this section, we present the results of our experimental study, which was conducted on two real-world datasets, namely CIFAR-10 and ImageNet[3], and one synthetic 2D dataset. We address all facets of a holistic evaluation metric, encompassing the fidelity, diversity, and novelty of the generated samples. In addition, we investigate computational efficiency. We use data examples and implementations of state-of-the-art evaluation metrics provided in (Stein et al., 2024; Jiralerspong et al., 2024). Unless otherwise stated, an equal number of 10000 training, test, and generated samples is utilized. We also emphasize that, in most of our experiments, we validate our method against FLD, which, to our best knowledge, is the only metric that considers all of the mentioned aspects (i.e., fidelity, diversity, and novelty) in one score. To this end, we apply an experimental setup from

---

[3]This choice is due to the fact that these datasets provide an explicit separation between training and test data examples.

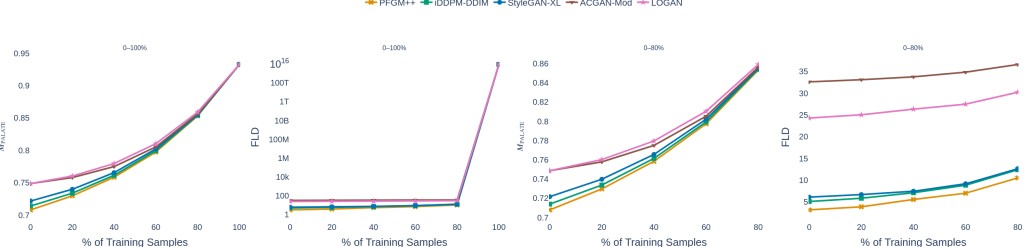

Figure 3: *Left:* $M_{\text{PALATE}}$ and FLD evaluated on the mixture of generated and training images from CIFAR-10, ranging from $0\%$ train (purely generated) to $100\%$ train (purely training). Since FLD eventually "blows up," its y-axis is plotted on a log scale. *Right:* Direct comparison within the range $[0\%, 80\%]$.

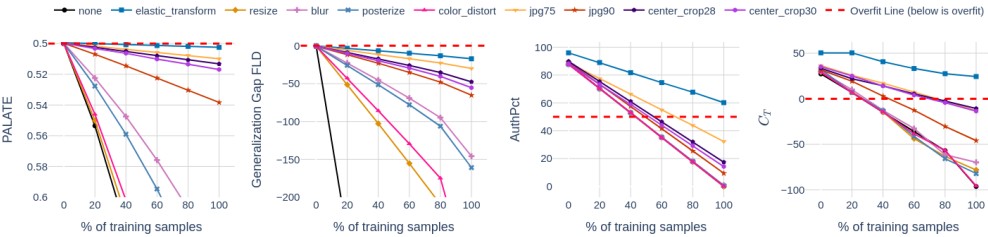

Figure 4: Capability of PALATE (our) to capture sample novelty, compared to different memorization metrics. The $y$-axis for our metric has been inverted for visual consistency.

(Jiralerspong et al., 2024). The source code can be found in the supplementary materials and will be made publicly available.

**Evaluation of DGMs**    In Table 1, we present a comparison of our approach with state-of-the-art evaluation metrics for a variety of DGMs on CIFAR-10 and ImageNet datasets. The results show that $M_{\text{PALATE}}$ provides a ranking similar to that of FLD. Furthermore, it is observed that models trained on ImageNet have a slightly higher tendency to overfit (PALATE $> 1/2$), which is consistent with negative Generalization Gap FLD scores.

**Sample Fidelity**    Figure 1 shows the behavior of $M_{\text{PALATE}}$ and FLD when different image distortions are applied to samples generated by PFGM++ trained on CIFAR-10. For the imperceptible transformations ("Posterize", "Light Blur", and "JPG90"), both metrics have slightly worse values compared to the baseline; however, they indicate that the samples are still better than those generated by StyleGAN-XL. On the other hand, for large transformations such as "Elastic Transform" or "Heavy Blur," both metrics evaluate the produced samples as worse than those produced by StyleGAN2-ada, which is the expected behavior. In summary, the negative impact on both metrics is proportional to the disturbance strength in both cases.

**Sample Diversity**    To assess the ability to accurately capture sample diversity, $M_{\text{PALATE}}$ was evaluated on the CIFAR-10 dataset with varying numbers of classes in the generated samples from different conditional generative models. For each class, 1000 samples were randomly selected and fixed to ensure that the same samples were used across all classes. This approach allows us to accurately measure how the metric responds to the addition of new classes, independent of variations in sample selection. To keep the total sample size constant while varying the number of classes, we adjusted the number of times each sample was duplicated based on the number of classes included. Specifically, when $C$ classes were included, 1000 pre-determined samples were selected from each class and duplicated equally for a total of 10000 samples. The duplication factor was determined by dividing 10000 by the number of samples selected, meaning that each sample was repeated $10/C$ times. This ensured a consistent sample size while systematically increasing class diversity.

In the other experiment, all classes were represented, but each generated sample was replicated an equal number of times to reach a total of 10000 samples. To achieve this, a set of 1000 samples was randomly selected and fixed for each class. Then, equal-sized subsets of each class were taken

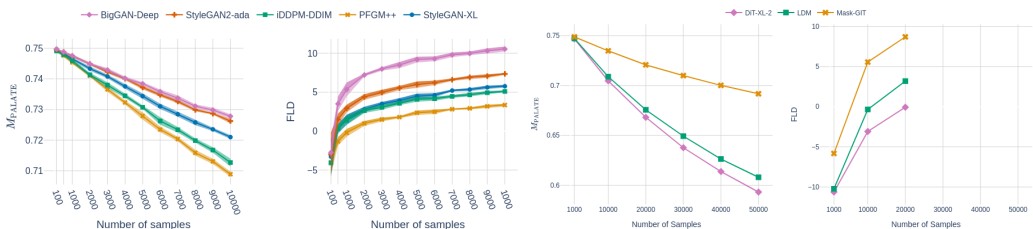

Figure 5: *Left*: Evaluation of $M_{\text{PALATE}}$ and FLD for different sample sizes on the CIFAR-10 dataset. *Right*: Evaluation of $M_{\text{PALATE}}$ and FLD for different sample sizes on the ImageNet dataset. The FLD plot is truncated at a sample size of 20000 due to its memory inefficiency on larger datasets.

and multiplied as necessary to achieve the desired total sample size. Precisely, for the number of $N$ unique samples per class, each sample was replicated $10000/(NC)$ times. This experimental design allowed for controlled variation in sample diversity while maintaining class balance.

As shown in Figure 2, $M_{\text{PALATE}}$ achieves comparable results to FLD in both scenarios. The metric scores obtained demonstrate a decreasing trend as either new classes are introduced or the number of unique samples per class increases, indicating that it adequately reflects the diversity variations in the generated samples.

**Sample Novelty**    To investigate the ability of our metric to capture sample novelty, we conducted an experiment in which samples generated by DGMs (trained on CIFAR-10) were progressively mixed with those from the training dataset. The results are presented in Figure 3. We can observe that values of both $M_{\text{PALATE}}$ and FLD increase (indicating overfitting) as more training samples are added. However, the increase in $M_{\text{PALATE}}$ is more gradual and smooth compared to the sharp spike observed in FLD, which experiences an explosive jump from $80\%$ to $100\%$ of the training samples. This suggests that $M_{\text{PALATE}}$ captures novelty more consistently, while FLD shows a more abrupt shift when the training dataset becomes complete.

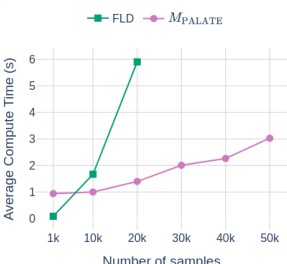

Figure 6: Computing time comparison for both metrics on ImageNet. The FLD plot is truncated at a sample size of 20000 due to its memory inefficiency on larger datasets.

Expanding on this, we apply various distortions to the mixture of images from PFGM++ and images from the training dataset. Within this setup, we compared PALATE with different memorization metrics, namely Generalization Gap FLD, AuthPct, and $C_T$ score. The results are shown in Figure 4. It can be observed that PALATE performs comparably to the other metrics, as its value increases with the increasing involvement of training samples in the evaluation, regardless of the distortions applied. However, it struggles with more severe distortions, such as elastic transform, a challenge shared by all of the metrics compared.

**Computational Efficiency and Stability**    We assess $M_{\text{PALATE}}$ and FLD on CIFAR-10 and ImageNet using various sample sizes, as illustrated in Figure 5. As the sample size increases, both metrics exhibit monotonic behavior. Despite neither plot reaching a plateau, the ranking of DGMs remains consistent across both metrics. However, the results obtained indicate the necessity of using larger sample sizes for both metrics, which warrants further investigation into computational efficiency. This is presented in Figure 6, where we compare the computation time of $M_{\text{PALATE}}$ and FLD on the ImageNet dataset, with experiments performed on the NVIDIA GeForce RTX 4090 GPU. As shown, $M_{\text{PALATE}}$ outperforms FLD in terms of computation time when the number of samples ranges from about 5000 to 20000. Additionally, FLD shows memory inefficiencies when processing 30000 samples or more, preventing us from computing it for such large datasets. This limitation is significant since FLD shows instability (or even negative values) for small sample sizes (see Figure 5). The efficiency of $M_{\text{PALATE}}$ is largely due to its reliance on matrix multiplications, which

are highly parallelizable and optimized in deep learning libraries such as TensorFlow, PyTorch, and JAX. Computational times were averaged over fifteen random seeds.

**Experiment on Synthetic Data** To further investigate the ability of $M_{\text{PALATE}}$ to capture model fit, an experiment was performed on synthetic 2D data. First, 2000 samples were generated from the mixture of three isotropic Gaussian distributions $\mathcal{N}(\cdot, I)$ centered at the vertices of an equilateral triangle with side length 3, which were then divided into the training and test datasets in a $50/50$ ratio. The process of training the model was simulated by sampling from KDEs computed with bandwidth $\sigma$ varying from $10^{-4}$ to $10^2$. For each value of $\sigma$, 1000 samples were generated and compared to the original data distribution using three evaluation metrics: FLD, FID, and $M_{\text{PALATE}}$.

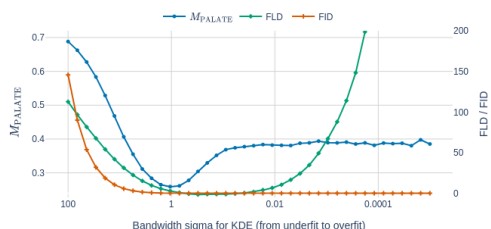

Figure 7: Evaluation of model fitting using $M_{\text{PALATE}}$ and FLD across varying values of $\sigma$. The results were averaged over 100 runs with different seeds.

As shown in Figure 7, sweeping through a range of $\sigma$, starting from high values that lead to underfitting and gradually decreasing to low values that lead to overfitting, it can be observed that $M_{\text{PALATE}}$ decreases as the model better captures the data distribution. FLD exhibits similar behavior, but reaches its minimum later. Both metrics reflect for overfitting by increasing their values. In contrast, while FID effectively tracks the "training" phase, it fails to adapt when the generated samples closely match the training data, making it unsuitable for comprehensive evaluation.

## 6  CONCLUSIONS

This work proposes PALATE, a novel enhancement to the evaluation of deep generative models grounded in the law of total expectation. It provides assessment sensitive to sample memorization and overfitting. By combining PALATE with an MMD baseline metric and leveraging DINOv2 embeddings, we obtain a holistic evaluation tool which accounts for fidelity, diversity, and novelty of generated samples while maintaining computational efficiency. Experiments conducted on the CIFAR-10 and ImageNet datasets demonstrate the ability of the proposed metric to reduce computational demands while preserving evaluation efficiency, which is comparable or even superior to that of state-of-the-art competitors.

**Limitations** The primary constraint of the proposed metric is limited range, attributable to minimal disparities between values of the baseline metric calculated for the train and test data samples. Additionally, our approach has not yet been evaluated beyond the domain of DGMs trained on image datasets. These limitations are considered potential avenues for future research endeavors.

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

## A BROADER IMPACTS

The objective of this work was to enhance the evaluation process of deep generative models. It is crucial to recognize that the implementation of generative modeling in real-world applications necessitates meticulous oversight to avert the intensification of societal biases embedded in the data. Moreover, it is anticipated that the findings of our study will exert an influence on subsequent research in related domains, particularly in the field of deepfake detection, which has witnessed a marked increase in interest due to advancements in deep generative models. The emergence of sophisticated techniques has led to a significant challenge in discerning authenticity from fabrication, underscoring the critical importance of evaluation in mitigating potential threats.

## B  GenAI Usage Disclosure

Generative AI software tools were used exclusively during the writing stage to edit and improve the clarity and quality of the existing manuscript text. No AI-generated content was used to produce novel research ideas, analyses, or results.

## C  Additional Theoretical Study

**Data-copying**  In (Meehan et al., 2020), the authors introduced the concept of data-copying, a form of overfitting that differs from previous work investigating over-representation (Heusel et al., 2017; Sajjadi et al., 2018). Intuitively, it refers to situations where the model distribution $p_G$ is closer to the train dataset than the real data distribution $p_X$ happens to be. Below we provide a precise definition, keeping notation introduced in the main paper.

**Definition 1** *Let $d\colon \mathcal{X} \to \mathbb{R}$ be a function that quantifies a squared Euclidean distance to the train dataset (calculated in a feature space). A given generative model is said to be data-copying the train dataset, if random draws from $d(Y)$ are systematically smaller then random draws from $d(X)$, i.e.,* $\mathbb{E}(\mathbb{1}_{d(Y)<d(X)}) > \frac{1}{2}$, *where $\mathbb{1}$ denotes a characteristic function.*

On the other hand, having done a separate test dataset for evaluation, like in our approach, we can modify condition from Definition 1 in order to take into account situations where the model distribution is closer to the train dataset than to the test dataset. This is a subject of the following definition.

**Definition 2** *A given generative model is said to be data-copying the train dataset, relative to the test dataset, if random draws from $Y$ are in average closer to random draws from $X$ conditioned on $X \in \mathcal{M}_{\text{train}}$ than to those conditioned on $X \in \mathcal{M}_{\text{test}}$, i.e.:*

$$\mathbb{E}(Z|X \in \mathcal{M}_{\text{train}}) < \mathbb{E}(Z|X \in \mathcal{M}_{\text{test}}). \tag{7}$$

It should be noted that generative models satisfying the above definition are exactly those for which we have the PALATE score greater than $a = \mathbb{P}(X \in \mathcal{M}_{\text{test}})$ (recall that we set $a = 1/2$ in our metric implementation). To see this, we need to prove the following equivalence:

$$E(Z|X \in M_{\text{train}}) < E(Z|X \in M_{\text{test}}) \Longleftrightarrow \text{PALATE}(M_{\text{base}}) > a. \tag{8}$$

"$\Longrightarrow$" Assuming $E(Z|X \in M_{\text{train}}) < E(Z|X \in M_{\text{test}})$, we obtain

$$\text{PALATE}(M_{\text{base}}) > \frac{aE(Z|X \in M_{\text{test}})}{aE(Z|X \in M_{\text{test}}) + (1-a)E(Z|X \in M_{\text{test}})} = \frac{aE(Z|X \in M_{\text{test}})}{E(Z|X \in M_{\text{test}})} = a. \tag{9}$$

"$\Longleftarrow$" If $\text{PALATE}(M_{\text{base}}) > a$, then

$$E(Z|X \in M_{\text{test}}) > aE(Z|X \in M_{\text{test}}) + (1-a)E(Z|X \in M_{\text{train}}), \tag{10}$$

which implies that

$$E(Z|X \in M_{\text{test}}) > E(Z|X \in M_{\text{train}}). \tag{11}$$

Consequently, we conclude that DGMs exhibiting a tendency to memorization are exactly those for which $\text{PALATE}(M_{\text{base}}) > a$.

**Proof of Theorem 1**  In general, the expectation of a random variable $Z$ over a sample space $\Omega$ is provided by the following formula:

$$\mathbb{E}(Z) = \int_{\Omega} Z(\omega)\, d\mathbb{P}(\omega), \tag{12}$$

where $\mathbb{P}$ is the probability measure on $\Omega$. Given the partition $\{A_1, \ldots, A_n\}$ of $\Omega$, we can decompose the above integral into a sum of integrals over each event $A_i$, i.e.:

$$\mathbb{E}(Z) = \int_{\Omega} Z(\omega)\, d\mathbb{P}(\omega) = \sum_{i=1}^{n} \int_{A_i} Z(\omega)\, d\mathbb{P}(\omega). \tag{13}$$

Then, since $\mathbb{P}(A_i) > 0$, we can express the integral over $A_i$ using the definition of conditional expectation:

$$\int_{A_i} Z(\omega)\, d\mathbb{P}(\omega) = \mathbb{E}(Z|A_i)\, \mathbb{P}(A_i). \tag{14}$$

Substituting this back into the sum in Equation (13) we obtain:

$$\mathbb{E}(Z) = \sum_{i=1}^{n} \int_{A_i} Z(\omega)\, d\mathbb{P}(\omega) = \sum_{i=1}^{n} \mathbb{E}(Z|A_i)\, \mathbb{P}(A_i), \tag{15}$$

which completes the proof.

**Proof of the Formula for $a = \mathbb{P}(X \in \mathcal{M}_{\textbf{test}})$** Consider any nontrivial measurable function $g\colon \mathcal{X} \to \mathbb{R}$. Applying the law of total expectation (see Theorem 1 in the main paper) to the random variable $g(X)$ and the partition of the sample space $\Omega = \{\omega \mid X(\omega) \in \mathcal{M}_{\text{test}}\} \cup \{\omega \mid X(\omega) \in \mathcal{M}_{\text{train}}\}$, we obtain:

$$\mathbb{E}(g(X)) = \mathbb{E}(g(X)|X \in \mathcal{M}_{\text{data}}) = a\, \mathbb{E}(g(X)|X \in \mathcal{M}_{\text{test}}) + (1-a)\, \mathbb{E}(g(X)|X \in \mathcal{M}_{\text{train}}). \tag{16}$$

Replacing all expectations in Equation (16) with their sample means yields:

$$\frac{1}{m+n}\left(\sum_{i=1}^{m} g(x_i^{\text{train}}) + \sum_{i=1}^{n} g(x_i^{\text{test}})\right) = \frac{1-a}{m}\sum_{i=1}^{m} g(x_i^{\text{train}}) + \frac{a}{n}\sum_{i=1}^{n} g(x_i^{\text{test}}), \tag{17}$$

From this, we can compute the hyperparameter $a$ as follows:

$$\begin{aligned}
a &= \frac{\frac{1}{m+n}\left(\sum_{i=1}^{m} g(x_i^{\text{train}}) + \sum_{i=1}^{n} g(x_i^{\text{test}})\right) - \frac{1}{m}\sum_{i=1}^{m} g(x_i^{\text{train}})}{\frac{1}{n}\sum_{i=1}^{n} g(x_i^{\text{test}}) - \frac{1}{m}\sum_{i=1}^{m} g(x_i^{\text{train}})} \\[2mm]
&= \frac{\frac{1}{m+n}\sum_{i=1}^{n} g(x_i^{\text{test}}) - \frac{n}{m(m+n)}\sum_{i=1}^{m} g(x_i^{\text{train}})}{\frac{1}{n}\sum_{i=1}^{n} g(x_i^{\text{test}}) - \frac{1}{m}\sum_{i=1}^{m} g(x_i^{\text{train}})} \\[2mm]
&= \frac{n}{m+n}\, \frac{\frac{1}{n}\sum_{i=1}^{n} g(x_i^{\text{test}}) - \frac{1}{m}\sum_{i=1}^{m} g(x_i^{\text{train}})}{\frac{1}{n}\sum_{i=1}^{n} g(x_i^{\text{test}}) - \frac{1}{m}\sum_{i=1}^{m} g(x_i^{\text{train}})} \\[2mm]
&= \frac{n}{m+n}.
\end{aligned} \tag{18}$$

It is noteworthy that the derived value of $a$ is independent of the given data samples and depends only on their sizes.

# D  Implementation Details

**Formulas for Calculating Metrics** We begin by providing complete formulas to compute the values of the PALATE and $M_{\text{PALATE}}$ metrics, based on available real and generated data samples $\mathrm{x}_{\text{train}} = \{x_1^{\text{train}}, \ldots, x_n^{\text{train}}\}$, $\mathrm{x}_{\text{test}} = \{x_1^{\text{test}}, \ldots, x_n^{\text{test}}\}$, and $\mathrm{y} = \{y_1, \ldots, y_n\}$. They display as follows:

$$\text{PALATE} = \frac{\bar{k}_{\mathrm{x}_{\text{test}}, \mathrm{x}_{\text{test}}} + \bar{k}_{\mathrm{y}, \mathrm{y}} - 2\bar{k}_{\mathrm{x}_{\text{test}}, \mathrm{y}}}{\bar{k}_{\mathrm{x}_{\text{test}}, \mathrm{x}_{\text{test}}} + \bar{k}_{\mathrm{y}, \mathrm{y}} - 2\bar{k}_{\mathrm{x}_{\text{test}}, \mathrm{y}} + \bar{k}_{\mathrm{x}_{\text{train}}, \mathrm{x}_{\text{train}}} + \bar{k}_{\mathrm{y}, \mathrm{y}} - 2\bar{k}_{\mathrm{x}_{\text{train}}, \mathrm{y}}}, \tag{19}$$

$$M_{\text{PALATE}} = \frac{1}{2}\frac{\bar{k}_{\mathrm{x}_{\text{test}}, \mathrm{x}_{\text{test}}} + \bar{k}_{\mathrm{y}, \mathrm{y}} - 2\bar{k}_{\mathrm{x}_{\text{test}}, \mathrm{y}}}{\bar{k}_{\mathrm{x}_{\text{test}}, \mathrm{x}_{\text{test}}} + \bar{k}_{\mathrm{y}, \mathrm{y}}} + \frac{1}{2}\,\text{PALATE}, \tag{20}$$

where $\bar{k}_{\cdot,\cdot}$ are respective $V$-statistics for kernel expectations, i.e.:

$$\bar{k}_{x_{\text{test}},x_{\text{test}}} = \frac{1}{n^2} \sum_{i,j=1}^{n} \exp(-\|x_i^{\text{test}} - x_j^{\text{test}}\|^2/(2\sigma^2)), \tag{21}$$

$$\bar{k}_{x_{\text{train}},x_{\text{train}}} = \frac{1}{n^2} \sum_{i,j=1}^{n} \exp(-\|x_i^{\text{train}} - x_j^{\text{train}}\|^2/(2\sigma^2)), \tag{22}$$

$$\bar{k}_{y,y} = \frac{1}{n^2} \sum_{i,j=1}^{n} \exp(-\|y_i - y_j\|^2/(2\sigma^2)), \tag{23}$$

$$\bar{k}_{x_{\text{test}},y} = \frac{1}{n^2} \sum_{i,j=1}^{n} \exp(-\|x_i^{\text{test}} - y_j\|^2/(2\sigma^2)), \tag{24}$$

$$\bar{k}_{x_{\text{train}},y} = \frac{1}{n^2} \sum_{i,j=1}^{n} \exp(-\|x_i^{\text{train}} - y_j\|^2/(2\sigma^2)), \tag{25}$$

and $\sigma$ is a dataset dependent constant, i.e., $\sigma = 10$ for the real world datasets and $\sigma = 1$ for the synthetic 2D dataset.

Table 2: List of torchvision functions with corresponding parameters for image transformations used in our experiments.

| Transformation | Python function | Arguments |
|---|---|---|
| Posterize | torchvision.transforms.functional.posterize | bits=5 |
| Light Blur | torchvision.transforms.GaussianBlur | kernel_size=5, sigma=0.5 |
| Heavy Blur | torchvision.transforms.GaussianBlur | kernel_size=5, sigma=1.4 |
| Center Crop 30 | torchvision.transforms.functional.center_crop torchvision.transforms.functional.pad | output_size=30 padding=1 |
| Center Crop 28 | torchvision.transforms.functional.center_crop torchvision.transforms.functional.pad | output_size=28 padding=2 |
| Color Distort | torchvision.transforms.ColorJitter | brightness=0.5, contrast=0.5, saturation=0.5, hue=0.5 |
| Elastic transform | torchvision.transforms.ElasticTransform | - |

**Algorithmic Specifics** The DMMD implementation uses a memory-efficient approach to calculate the maximum mean discrepancy (MMD) between two sets of embeddings by splitting the data into smaller parts with a block size of 1000. Experimentally, this block size has been found to offer the fastest performance by balancing memory usage and computational speed effectively. Instead of creating full kernel matrices, DMMD processes the data in chunks of 1000 samples at a time. For each chunk, it computes parts of the kernel matrix separately and then combines the results, keeping memory requirements low. Additionally, the @jax.jit decorator helps boost performance by compiling the function with XLA (accelerated linear algebra). This allows the code to execute faster on GPUs and TPUs by optimizing operations and running them in parallel. As a result, this combination of the experimentally chosen block size and JIT compilation makes DMMD both memory-efficient and fast, enabling it to handle large datasets effectively.

**Transformations** For the experiments with image transformations we used popular torchvision library (maintainers & contributors, 2016). The exact function names and corresponding arguments are listed in Table 2.

