# OpenReview forum: "PALATE: Peculiar Application of the Law of Total Expectation to Enhance the Evaluation of Deep Generative Models"
_ICLR.cc/2026/Conference — Submitted to ICLR 2026_

### Official Review · Reviewer_cq4n · 2025-11-01

**Soundness:** 3
**Presentation:** 4
**Contribution:** 3
**Rating:** 6
**Confidence:** 4

**Summary:**

This paper introduces an evaluation framework that contrasts training data points with generated samples by leveraging the law of total expectation. Concretely, the authors define a ratio-based score (PALATE) that decomposes an expectation-based baseline metric into contributions conditioned on whether a real sample comes from the train or the held-out set, so that the score rises when generations are closer to the training set (a signal of memorization) than to the test set. Because this ratio alone can have limited dynamic range in some regimes, they further propose a weighted metric (MPALATE) that takes a convex combination of the classical baseline and the new ratio—thereby balancing fidelity/diversity sensitivity (from the baseline) with novelty/memorization sensitivity (from PALATE) and stabilizing behavior across datasets. The framework is instantiated with MMD computed on pretrained feature embeddings (RBF kernel), and evaluated on standard image benchmarks; results are compared against FLD (a holistic but more resource-hungry baseline) and FID (fidelity-oriented). Empirically, the method aims to retain competitive fidelity/diversity tracking while improving memorization detection and offering better computational scalability than FLD.

**Strengths:**

- **Well-written with a fair experimental setup.** The paper is clear, and the experiments are aligned closely with **FLD** (datasets, budgets, protocol), which makes the comparison credible.
- **Elegant formulation via the law of total expectation.** The construction cleanly injects a **novelty/memorization** signal into an expectation-based baseline without changing the two-sample structure, so it’s easy to adopt.
- **Computational efficiency is convincingly demonstrated.** By avoiding KDE and using **V-statistic MMD** with blockwise kernel computations, the method scales to larger sample sizes where FLD becomes cumbersome. Attention to feature maps and implementation details (block sizes, batching) is practical and goes beyond what comparable work typically reports.
- **Plug-and-play with existing pipelines.** Since the base quantity is an expectation, the approach can wrap standard metrics with minimal engineering.

**Weaknesses:**

- **Arbitrary design choices with limited sensitivity analysis.** Several decisions feel ad hoc (choice of **MMD**, kernel family/bandwidth, feature extractor, the convex weight **α**, score scaling).
- **Lack of non-image modalities.** Recent work extends distributional metrics from **images to text**; adding text experiments (even small-scale) using the framework with textual baselines such as **MAUVE** or precision/recall for text models would strengthen the claim of generality [1, 2, 3].
- **No disentanglement of quality vs. diversity in the chosen base metric.** MMD is a single-number distance and cannot separate fidelity from coverage. However, **precision/recall**  estimated as in [4] or [5] can be written as **f-divergences** [6,7], hence as **expectations of a function**. They should fit your framework as naturally as MMD, with the key advantage of **disentangling quality and diversity**. Exploring **PALATE-Precision** and **PALATE-Recall** (and their weighted variants) could reveal more nuanced behaviors.


---
[1] Pillutla et al., **MAUVE: Measuring the Gap Between Neural Text and Human Text using Divergence Frontiers**, NeurIPS 34.

[2] Le Bronnec et al., **Exploring Precision and Recall to assess the quality and diversity of LLMs**, ACL.

[3] Pimentel et al., **On the usefulness of embeddings, clusters and strings for text generator evaluation**, ICLR 2023.

[4] Kynkäänniemi et al., **Improved Precision and Recall Metric for Assessing Generative Models**, NeurIPS 2019.

[5] Kim et al., **TopP&R: Robust Support Estimation Approach for Evaluating Fidelity and Diversity in Generative Models**, NeurIPS 2025.

[6] Siry et al., **On the Theoretical Equivalence of Several Trade-Off Curves Assessing Statistical Proximity**, JMLR 2023.

[7] Verine et al., **Precision-Recall Divergence Optimization for Generative Modeling with GANs and Normalizing Flows**, NeurIPS 37.

**Questions:**

- Why prioritize **MMD** over alternative expectation-form distances (e.g., energy distance, discriminators)?
- Since **precision/recall** families can be written as **f-divergences** (hence expectations), could you implement **PALATE-Precision** and **PALATE-Recall**? This would naturally separate fidelity from coverage and might provide clearer guidance to practitioners.
- Is the computational gain is only over **FLD**, or does it also hold against other expectation-based metrics such as **FID** or **KID**, **PRDC**?

**Details Of Ethics Concerns:**

--

---

> ### Author Response · Authors · 2025-11-21
>
> We thank the reviewer for recognizing the soundness, elegant formulation, clear presentation, and fair experimental setup of our approach, as well as its computational efficiency and ease of integration with existing methods. We appreciate the constructive feedback and have addressed all questions and concerns raised below. Should the reviewer find our responses satisfactory, we kindly request reconsideration of the score. We remain available to provide any further clarifications as needed.
>
> * *Why prioritize MMD over alternative expectation-form distances (e.g., energy distance, discriminators)?
> Since precision/recall families can be written as f-divergences (hence expectations), could you implement PALATE-Precision and PALATE-Recall? This would naturally separate fidelity from coverage and might provide clearer guidance to practitioners.*
>
> In fact, the choice of hyperparameters was not straightforward. Despite theoretical analysis, we had to examine various experimental configurations to establish a final setup. On the other hand, we do not abandon searching for better solutions; we treat this as work for the future.
>
> Though MMD-based metrics are well-suited candidates as baseline metrics due to their definition via the expectation of a random variable, any evaluation metric that satisfies assumption (A3) may be considered. In fact, we can treat every metric as satisfying this assumption if we consider $Z$ to be a random variable with a Dirac delta probability measure centered at the obtained metric value. Note that, in this case, we have $E(Z|X\in M_\text{test})=M_\text{base}$. This concept looks somewhat tricky, but it allows for the use of FID as a baseline metric. We made an attempt to do just that. However, we were unable to establish a reasonable scaling method. This was due to the difficulty of finding an upper bound for FID across all possible models.
>
> Concerning MMD-based metrics, despite DMMD, we tested CMMD (with a CLIP embedding instead of a DINOv2 embedding), KID, and energy distance (though it can be considered as equivalent to MMD [3]); however, the results were not promising. In fact, our final decision to use DINOv2 was motivated by CLIP's lack of fine-grained recognition capabilities [2]. We consider this ability crucial for detecting memorization, as explained in Section 4 (see the lines 250-259 in the original version of the paper). On the other hand, KID uses a non-characteristic polynomial kernel, unlike DMMD, which uses a Gaussian RBF characteristic kernel. This results in non-distinguishing distributions with the same first three moments (note that distinguishing subtle object features is also important when trying to detect memorization).
>
> However, we believe that our future work will provide interesting alternatives for choosing a baseline metric.
>
> * *Lack of non-image modalities. Recent work extends distributional metrics from images to text; adding text experiments (even small-scale) using the framework with textual baselines such as MAUVE or precision/recall for text models would strengthen the claim of generality [1, 2, 3].*
>
> This is a great suggestion! Although we have not yet tested our metric beyond image data, we believe that, due to its theoretical foundations, the general approach will be applicable to other data modalities. We consider this an important direction for future work. For instance, one of our upcoming projects is to extend the PALATE approach to tabular datasets.
>
> * *No disentanglement of quality vs. diversity in the chosen base metric. MMD is a single-number distance and cannot separate fidelity from coverage. However, precision/recall estimated as in [4] or [5] can be written as f-divergences [6,7], hence as expectations of a function. They should fit your framework as naturally as MMD, with the key advantage of disentangling quality and diversity. Exploring PALATE-Precision and PALATE-Recall (and their weighted variants) could reveal more nuanced behaviors.*
>
> In fact, we explored using the PALATE enhancement to improve precision and recall metrics, which would result in two separate evaluation metrics: one assessing fidelity/memorization and the other assessing diversity/memorization. However, since our primary goal was to define a holistic, single-value metric that simultaneously captures fidelity, diversity, and memorization, we decided not to pursue this direction further. (Nonetheless, following the reviewer’s suggestion, this may be a worthwhile avenue to revisit in future work.) We believe the key advantage of a single-value metric is its ability to quickly distinguish between modern deep generative models. Notably, recent diffusion models have been observed to replicate training data (see, for example, [1]), underscoring the importance of a holistic evaluation that incorporates this aspect.

---

> > ### Author Response · Authors · 2025-11-21
> > **Official Comment by Authors (continued)**
> >
> > * *Is the computational gain is only over FLD, or does it also hold against other expectation-based metrics such as FID or KID, PRDC?*
> >
> > The computational advantage of $M_\text{PALATE}$ is relative solely to FLD, another holistic metric and a direct state-of-the-art competitor. FLD’s complexity arises from its use of kernel density estimation (KDE) with a bandwidth matrix that must be optimized. This optimization requires storing and manipulating a distance matrix of size $n\times n$, where $n$ is the number of samples, which causes memory inefficiency when processing datasets of 30,000 samples or more, as demonstrated in our computational efficiency experiments. Additionally, FLD necessitates estimating a data-dependent constant, which can occasionally produce negative metric values [4].
> >
> > *References*
> >
> > [1] Somepalli, Gowthami, et al. Understanding and mitigating copying in diffusion models. Advances in Neural Information Processing Systems 36 (2023): 47783-47803.
> >
> > [2] Bianchi, Lorenzo, et al. Is CLIP the main roadblock for fine-grained open-world perception?. 2024 International Conference on Content-Based Multimedia Indexing (CBMI). IEEE, 2024.
> >
> > [3] Sejdinovic, Dino, et al. "Equivalence of distance-based and RKHS-based statistics in hypothesis testing." The annals of statistics (2013): 2263-2291.
> >
> > [4] Jiralerspong, Marco, et al. Feature likelihood divergence: evaluating the generalization of generative models using samples. Advances in Neural Information Processing Systems 36 (2023): 33095-33119.

---

### Official Review · Reviewer_yoJx · 2025-11-02

**Soundness:** 2
**Presentation:** 3
**Contribution:** 2
**Rating:** 2
**Confidence:** 4

**Summary:**

The authors propose PALATE, a new approach to evaluating deep generative models that builds upon the law of total expectation and prior work such as the FLD. A main idea is to leverage a total expectation that is based on a combination of an expectation on the testing data and an expectation on the training data to distinguish models that are merely memorizing.

**Strengths:**

Overall, I find the clarity and quality aspects to be strengths of this paper. I discuss significance and originality more in the weaknesses section.

Clarity: The clarity and accessibility of writing is a strength of this paper. The exposition is simple and clear, and the writing is easy to follow to a general ML audience. Overall it was a pleasant read.
Quality: I comment on the quality of 1. the math/theory portion of the paper 2. the empirical portion of the paper.
The quality of the mathematical expressions/technical portion is sound to the best of my knowledge. There is minimal theoretical developments in the paper, but this is not a problem since this paper's main contribution is methodological. The experimental section is structured reasonably and appropriate comparisons to competitors are made.
Significance: The problem of deep generative model evaluation is a significant and timely one. While the topic is significant, whether the paper's contribution is significant is discussed in the next section.
Originality:  I comment on the originality of the paper's contribution in the next section.

**Weaknesses:**

On the significance and originality front, I have the following comments and questions.

1. The core idea of PALATE depends on a split of the data into training and testing portion, and then comparing the relative contributions of each component to determine and balance whether the data is memorizing/overfitted to the training data and whether it faithfully generates points similar to the test data.

However, this core idea of train test split has been studied and proposed in the machine learning and statistics literature for many decades. In this sense, I find the core idea less original than the paper's description and less significant of a conceptual advance. I also think that there is too little discussion of the relationship with traditional data splitting/cross validation ideas.

2. the PALATE functional form essentially compares a ratio between an expected "loss" on the test data and an expected loss on the entire data. While this functional form is, to the best of my knowledge, new in the deep generative modeling context, I question whether this is a sufficiently significant/original contribution for a venue like ICLR.

3. Simplicity if a virture, and the fact that the paper uses relatively elementary/simple mathematical tools is a good thing, so my comment below is not a criticism against the technical depth of the paper per se. However, I find the paper's emphasis on the law of total expectation quite confusing....the law of total expectation is such an elementary/fundamental technique that I find emphasizing that no different from emphasizing other basic things like the laws of arithmetic. I find it difficult to see the significant advancement when such an elementary tool is applied in such a straightforward manner.

**Questions:**

1. The authors reference Weiss et al for the law of total expectation. However, I don't find such a citation satisfactory. If the authors want to emphasize the law of total expectation, then proper reference to this law is warranted. Candidate references that might suffice include the original work of Kolmogorov and the classical expositions by Feller.

2. I do not find the application of the law of total expectation "peculiar" in any way. It is just a straightforward mathematical manipulation. Could the authors clarify what they mean by peculiar?

---

> ### Author Response · Authors · 2025-11-21
>
> We thank the reviewer for recognizing the theoretical soundness, clarity of presentation, and the appropriateness of the experimental study in our work, as well as the significance of the problem addressed. We appreciate the constructive feedback and have addressed all questions and concerns raised below. Should the reviewer find our responses satisfactory, we kindly request reconsideration of the score. We remain available to provide any further clarifications if needed.
>
> * *The core idea of PALATE depends on a split of the data into training and testing portion, and then comparing the relative contributions of each component to determine and balance whether the data is memorizing/overfitted to the training data and whether it faithfully generates points similar to the test data. However, this core idea of train test split has been studied and proposed in the machine learning and statistics literature for many decades. In this sense, I find the core idea less original than the paper's description and less significant of a conceptual advance. I also think that there is too little discussion of the relationship with traditional data splitting/cross validation ideas.*
>
> The PALATE approach does not introduce the concept of data splitting itself but rather incorporates the train-test split within a comprehensive evaluation framework, similar to FLD, which (to the best of our knowledge) was the first metric to utilize this approach for jointly assessing fidelity, diversity, and memorization. Unlike cross-validation, which is a training technique used to estimate model generalization during learning [6], our method employs a train-test split solely for evaluation purposes of deep generative models. Traditional evaluation metrics often consider the entire dataset without explicitly distinguishing between training and testing subsets, thereby limiting the assessment to fidelity and diversity while precluding the measurement of memorization and overfitting. By explicitly integrating the split into the evaluation metric, the resulting holistic metric $M_\text{PALATE}$ explicitly quantifies how closely generated samples resemble training data versus novel test data, addressing a significant gap in classical frameworks.
>
>
> In summary, the PALATE approach builds upon the concept of data splitting but does not claim novelty in the splitting itself. Instead, it applies the split within a principled, post-learning evaluation metric that captures memorization, an aspect that classical whole-dataset evaluations do not directly address. Following the reviewer's suggestion, in the revised version we have incorporated the above discussion into Section 4.
>
> * *the PALATE functional form essentially compares a ratio between an expected "loss" on the test data and an expected loss on the entire data. While this functional form is, to the best of my knowledge, new in the deep generative modeling context, I question whether this is a sufficiently significant/original contribution for a venue like ICLR.*
>
> We agree that, in practice, our proposed metrics, $\text{PALATE}$ and $M_\text{PALATE}$, essentially involve applying direct formulas to the values of a baseline metric, DMMD, a variant of CMMD [2], but using DINOv2 representations instead of CLIP, which are calculated using training and testing datasets. However, we do not consider this to be a weakness. Let us briefly explain (in a few points) why.
>
> (1) These formulas were not designed in a simple combinatorial way. Rather, they were derived from a formal theoretical analysis (under the Manifold Hypothesis [3]) that involved the feature distributions of real and generated data.
>
> (2) The simplicity of the formulas is crucial for computational efficiency, a key advantage of our approach. For comparison, note that FLD, the most important state-of-the-art competitor of $M_\text{PALATE}$, suffers from a complicated definition that employs KDE with a bandwidth matrix that must be additionally optimized. This optimization procedure uses a stored distance matrix of dimension $n\times n$, where $n$ is the sample size. This leads to memory inefficiencies when processing 30,000 samples or more (see the results of our experiments regarding computational efficiency). Furthermore, calculating FLD requires estimating a data-dependent constant, which can result in negative metric values [1]. We believe that, in light of this, the simplicity of the definition of our metric is an advantage.
>
> (3) The choice of hyperparameters was not straightforward. Despite theoretical analysis, we had to examine various experimental configurations to establish a final setup. On the other hand, we do not abandon searching for better solutions; we treat this as work for the future.

---

> > ### Author Response · Authors · 2025-11-21
> > **Official Comment by Authors (continued)**
> >
> > * *Simplicity if a virture, and the fact that the paper uses relatively elementary/simple mathematical tools is a good thing, so my comment below is not a criticism against the technical depth of the paper per se. However, I find the paper's emphasis on the law of total expectation quite confusing....the law of total expectation is such an elementary/fundamental technique that I find emphasizing that no different from emphasizing other basic things like the laws of arithmetic. I find it difficult to see the significant advancement when such an elementary tool is applied in such a straightforward manner.*
> >
> > We appreciate the reviewer’s recognition of the simplicity of our approach as a strength. We understand and respect the reviewer’s perspective regarding the law of total expectation as a fundamental tool, similar to the laws of arithmetic. However, we kindly ask the reviewer to consider that the law of total expectation may not be as universally familiar, especially within the broader machine learning community, as basic arithmetic laws are. (This certainly contrasts with its recognition in the mathematical and statistical communities.) Moreover, while many machine learning researchers are aware of the law of total expectation, we hypothesize that only a smaller, specialized subset, namely theoreticians and experts in statistical learning, are likely to provide a precise statement of it on demand.
> >
> > * *The authors reference Weiss et al for the law of total expectation. However, I don't find such a citation satisfactory. If the authors want to emphasize the law of total expectation, then proper reference to this law is warranted. Candidate references that might suffice include the original work of Kolmogorov and the classical expositions by Feller.*
> >
> > We sincerely thank the reviewer for this comment. In response, in the revised version we have exchanged the citations to [4,5].
> >
> > * *I do not find the application of the law of total expectation "peculiar" in any way. It is just a straightforward mathematical manipulation. Could the authors clarify what they mean by peculiar?*
> >
> > We termed the use of the law of total expectation “peculiar” because this standard probability law was in an unconventional manner to decompose the evaluation metric. While mathematically straightforward, this approach represents an uncommon and elegant adaptation of a classical principle, tailored specifically for generative model evaluation, enabling holistic detection of memorization and generalization. However, we are open to any suggestions for alternative wording, preferably while maintaining the acronym "PALATE".
> >
> > *References*
> >
> > [1] Jiralerspong, Marco, et al. Feature likelihood divergence: evaluating the generalization of generative models using samples. Advances in Neural Information Processing Systems 36 (2023): 33095-33119.
> >
> > [2] Jayasumana, Sadeep, et al. Rethinking fid: Towards a better evaluation metric for image generation. Proceedings of the IEEE/CVF Conference on Computer Vision and Pattern Recognition. 2024.
> >
> > [3] Loaiza-Ganem, Gabriel, et al, Deep Generative Models through the Lens of the Manifold Hypothesis: A Survey and New Connections. Transactions of Machine Learning Research, 2024.
> >
> > [4] Kolmogorov, Andrei Nikolaevich. Foundations of the theory of probability. (1933).
> >
> > [5] Feller, William. An Introduction to Probability Theory and Its Applications, Vol. 1, 3rd Edition, Wiley, 1968.
> >
> > [6] Hastie, Trevor, Robert Tibshirani, and Jerome Friedman. The Elements of Statistical Learning: Data Mining, Inference, and Prediction. Springer Science & Business Media, 2009.

---

### Official Review · Reviewer_fG8R · 2025-11-02

**Soundness:** 3
**Presentation:** 2
**Contribution:** 2
**Rating:** 2
**Confidence:** 4

**Summary:**

The paper addresses the challenge of evaluating generative models by enhancing the Feature Likelihood Divergence (FLD) metric. The authors use the law of total expectation (tower rule) to reformulate the underlying evaluation process. They provide numerical results on CIFAR-10 and ImageNet (256×256) datasets generated by various generative models.

**Strengths:**

- The paper addresses an important problem in evaluating generative models and tries to tackle the computational issues of FLD.

- The paper uses DINOv2 feature embeddings, which have been shown to provide more reliable representations.

**Weaknesses:**

My major concern is the novelty of the contribution. The authors apply the law of total expectation (as explained in Theorem 1 and proved in the Appendix) to improve FLD. From a theoretical perspective, this improvement is incremental and does not address a substantial challenge. From an experimental perspective, most experiments are conducted using the CIFAR-10 dataset, which may not adequately demonstrate scalability or generalization to larger datasets. Below are more detailed versions of my concerns:

- Theoretical Basis: Using the law of total expectation gives a clear mathematical explanation, but the idea itself is mostly a reframing of existing metrics rather than a truly new approach.

- Feature Embedding: Using DINOv2 as the feature embedding is not original; this was proposed in [1], where the authors extensively studied embeddings and suggested that DINOv2 is a better alternative.

- Figure 4: Although the M$_{PALATE}$ score computation is faster, the score does not converge even with 10,000 samples, while FLD appears to converge with 8,000. This raises the question of what the sample complexity of your proposed score is. Do you have theoretical or experimental results for the convergence of your score?

- Also, the related work section could be improved by including more recent evaluation metrics. For example, Density/Coverage [2], Vendi score [3], FKEA [4], and RKE score [5] for fidelity/diversity evaluation, and FINC [6] for assessing novelty.

---

[1] Stein et. al, "Exposing flaws of generative model evaluation metrics and their unfair treatment of diffusion models", NeurIPS 2023

[2] Naeem, M., et al., “Reliable Fidelity and Diversity Metrics for Generative Models”, NeurIPS 2020.

[3] D. Friedman and A. B. Dieng, “The Vendi Score: A Diversity Evaluation Metric for Machine Learning.”, TMLR 2023

[4] Ospanov et al., “Towards a scalable reference-free evaluation of generative models”, NeurIPS 2024.

[5] Jalali et al., “An information-theoretic evaluation of generative models in learning multi-modal distributions”, NeurIPS 2023.

[6] Zhang et al., “Unveiling Differences in Generative Models: A Scalable Differential Clustering Approach”, CVPR 2025.

**Questions:**

- Figure 1: Which PFGM++ samples were used? Are these CIFAR-10 samples?

- Figure 2: Scores were reported for 1, 2, 5, 10 or 100, 200, 500, 1000 classes, which sometimes do not follow the expected trend. Is there a specific reason for this?

- Figure 3: Why are comparisons limited to AuthPCT and C_$T$? Why not include other baselines mentioned in the related work, such as KEN score and Rarity score?

- Figure 4: Scores were reported on CIFAR-10, while computation time is reported on the ImageNet dataset, which is inconsistent. Is there a specific reason for this? (I noticed that part of Figure 7 shows ImageNet, but the inconsistency remains.)

- Figure 5: There is a large gap between 80% and 100% for FLD, suggesting that FLD may be sensitive when all samples are from the training set. Can results be provided for the range [0, 90]% so that the results are more comparable?

---

> ### Author Response · Authors · 2025-11-21
>
> We thank the reviewer for recognizing the soundness and computational aspect of our approach, as well as the importance of the addressed problem and the appropriate choice of feature extractor. We appreciate the constructive feedback and address all questions and concerns below. If the reviewer finds our responses satisfactory, we kindly request reconsideration of the score. We remain available to provide any further clarifications if needed.
>
> * *My major concern is the novelty of the contribution. The authors apply the law of total expectation (as explained in Theorem 1 and proved in the Appendix) to improve FLD. From a theoretical perspective, this improvement is incremental and does not address a substantial challenge. Theoretical Basis: Using the law of total expectation gives a clear mathematical explanation, but the idea itself is mostly a reframing of existing metrics rather than a truly new approach.*
>
> We agree that, in practice, our proposed metrics, $\text{PALATE}$ and $M_\text{PALATE}$, essentially involve applying direct formulas to the values of a baseline metric, DMMD, a variant of CMMD [2], but using DINOv2 representations instead of CLIP, which are calculated using training and testing datasets. However, we do not consider this to be a weakness. Let us briefly explain (in a few points) why.
>
> (1) These formulas were not designed in a simple combinatorial way. Rather, they were derived from a formal theoretical analysis (under the Manifold Hypothesis [3]) that involved the feature distributions of real and generated data.
>
> (2) The simplicity of the formulas is crucial for computational efficiency, a key advantage of our approach. For comparison, note that FLD, the most important state-of-the-art competitor of $M_\text{PALATE}$, suffers from a complicated definition that employs KDE with a bandwidth matrix that must be additionally optimized. This optimization procedure uses a stored distance matrix of dimension $n\times n$, where $n$ is the sample size. This leads to memory inefficiencies when processing 30,000 samples or more (see the results of our experiments regarding computational efficiency). Furthermore, calculating FLD requires estimating a data-dependent constant, which can result in negative metric values [4]. We believe that, in light of this, the simplicity of the definition of our metric is an advantage.
>
> (3) The choice of hyperparameters was not straightforward. Despite theoretical analysis, we had to examine various experimental configurations to establish a final setup. On the other hand, we do not abandon searching for better solutions; we treat this as work for the future.
>
> * *From an experimental perspective, most experiments are conducted using the CIFAR-10 dataset, which may not adequately demonstrate scalability or generalization to larger datasets.*
>
>
> Our experimental setup closely follows that of the reference paper [4], where the authors introduced Feature Likelihood Divergence (FLD) and Generalization Gap FLD, direct competitors to $M_\text{PALATE}$ and $\text{PALATE}$. In fact, we also assess our metrics on ImageNet for model evaluation and computational efficiency (note that computational efficiency is an area in which $M_\text{PALATE}$ has a clear advantage over FLD, see the right side of Figure 4 in the original version of the paper). The experiments addressing metric fidelity, diversity, and novelty conducted on CIFAR-10 are almost identical to those presented in [4] (compare our Figures 1, 2, and 3 with Figures 5, 6, and 7 in [4]).
>
> * *Feature Embedding: Using DINOv2 as the feature embedding is not original; this was proposed in [1], where the authors extensively studied embeddings and suggested that DINOv2 is a better alternative.*
>
> We use DINOv2 representations for the baseline metric DMMD as they better capture global structure and salient objects, thereby improving perceptual alignment. While using DINOv2 embeddings was studied in [1], it has not been explored in the context of a kernel-based metric. Note that DMMD, like CMMD [2], is a variant of an MMD-based metric with a characteristic RBF kernel, but it employs DINOv2 embeddings instead of CLIP.

---

> > ### Author Response · Authors · 2025-11-21
> > **Official Comment by Authors (continued)**
> >
> > * *Figure 4: Although the M score computation is faster, the score does not converge even with 10,000 samples, while FLD appears to converge with 8,000. This raises the question of what the sample complexity of your proposed score is. Do you have theoretical or experimental results for the convergence of your score?*
> >
> > We appreciate the reviewer’s insightful concern regarding the convergence of our metric. Theoretically, $M_\text{PALATE}$ is fundamentally constructed from expectations estimated by V-statistics, which are unbiased, consistent estimators known to converge according to the law of large numbers. The metric itself results from simple arithmetic manipulations, specifically, a weighted ratio of these empirical expectations. Therefore, $M_\text{PALATE}$ should inherit consistent convergence properties of the underlying kernel-based measures. Empirically, $M_\text{PALATE}$ demonstrates stable, monotonic convergence trends with increasing sample sizes, with model rankings stabilizing beyond a few thousand samples.
> >
> > * *Also, the related work section could be improved by including more recent evaluation metrics. For example, Density/Coverage [2], Vendi score [3], FKEA [4], and RKE score [5] for fidelity/diversity evaluation, and FINC [6] for assessing novelty.*
> >
> > We have incorporated the reviewer’s suggestion to include the proposed metrics in the related work section, utilizing the extended page limit available for the revised version.
> >
> > * *Figure 1: Which PFGM++ samples were used? Are these CIFAR-10 samples?*
> >
> > Yes, these are CIFAR-10 samples. We indicated this in the revised version.
> >
> > * *Figure 2: Scores were reported for 1, 2, 5, 10 or 100, 200, 500, 1000 classes, which sometimes do not follow the expected trend. Is there a specific reason for this?*
> >
> > In fact, 1, 2, 5, and 10 represent the number of classes, while 100, 200, 500, and 1000 correspond to the number of unique samples per class included in the evaluation. In both cases, the scores decrease, which aligns with the expected trend. The differences in the rate of decrease likely result from unequal variations between classes, which are inherently difficult to quantify.
> >
> > * *Figure 3: Why are comparisons limited to AuthPCT and C_? Why not include other baselines mentioned in the related work, such as KEN score and Rarity score?*
> >
> > The reason for this is that our experimental setup closely follows that of the reference paper [4]. For a direct comparison, see our Figure 3 alongside Figure 7 in [4].
> >
> > * *Figure 4: Scores were reported on CIFAR-10, while computation time is reported on the ImageNet dataset, which is inconsistent. Is there a specific reason for this? (I noticed that part of Figure 7 shows ImageNet, but the inconsistency remains.)*
> >
> > To save space, Figure 4 (in the original version of the paper) combined the results of two independent experiments. In the revised version, taking advantage of the extended page limit, we have reorganized Figures 4 and 7 into a more consistent and clear layout.
> >
> > * *Figure 5: There is a large gap between 80% and 100% for FLD, suggesting that FLD may be sensitive when all samples are from the training set. Can results be provided for the range [0, 90]% so that the results are more comparable?*
> >
> > Indeed, in Figure 5 we aimed to demonstrate an abrupt shift in the FLD score when the training dataset becomes complete. However, following the reviewer’s suggestion, in the revised version we also included a separate comparison within the range [0%, 80%].
> >
> > *References*
> >
> > [1] Stein, George, et al. Exposing flaws of generative model evaluation metrics and their unfair treatment of diffusion models. Advances in Neural Information Processing Systems 36 (2023): 3732-3784.
> >
> > [2] Jayasumana, Sadeep, et al. Rethinking fid: Towards a better evaluation metric for image generation. Proceedings of the IEEE/CVF Conference on Computer Vision and Pattern Recognition. 2024.
> >
> > [3] Loaiza-Ganem, Gabriel, et al, Deep Generative Models through the Lens of the Manifold Hypothesis: A Survey and New Connections. Transactions of Machine Learning Research, 2024.
> >
> > [4] Jiralerspong, Marco, et al. Feature likelihood divergence: evaluating the generalization of generative models using samples. Advances in Neural Information Processing Systems 36 (2023): 33095-33119.

---

### Official Review · Reviewer_zDZj · 2025-11-03

**Soundness:** 1
**Presentation:** 1
**Contribution:** 1
**Rating:** 0
**Confidence:** 3

**Summary:**

The premise of this paper makes little sense. See attached questions. Unless it is revised to meet a basic scientific standard (precise definitions and sensible justifications behind the modeling choices) I won't be able to assess this manuscript.

**Strengths:**

Until the premise of this paper is qualified I cannot assess its strenghts.

**Weaknesses:**

Until the premise of this paper is qualified I cannot assess its weaknesses.

typo: In line 209 the term P(Z | X in M_test) does not parse. Z is a random variable.

**Questions:**

Line 97: "we have access to two distinct collections of real data: a train dataset and a test dataset, with the former dedicated for training and the latter for evaluation purposes exclusively. These data can be considered independently drawn from a random variable X acting on a given multidimensional Euclidean data space."

Line 190: "The following assumptions underpin our approach: (A1) the samples of training data which are selected for evaluation, are
contained in two non-trivial disjoint parts1 Mtrain and Mtest of the manifold of data M, respectively."

These two assumptions are inconsistent. If the training and test datasets are sampled from the same dataset, how can they be contained in disjoint parts of whatever manifold you are talking about? What is the DEFINITION of Mtrain and Mtest?

Line 198: "note that any such Z must implicitly depend on the random variables X and Y —see Equation (5) for an example." No it must not. The random variable Z that is identically equal to zero satisfies E[Z] = 0 but does not "implicitly depend" on any X and Y. There are myriad examples of random variables that have mean zero but do not "implicitly depend" on your X and Y.

---

> ### Author Response · Authors · 2025-11-21
>
> We respectfully disagree with the reviewer’s opinion. We believe it stems from a misreading or misunderstanding of our approach; therefore, we provide a detailed explanation below, addressing the concerns raised in the official review. We hope this clarification will help resolve the reviewer’s concerns and lead to a reconsideration of the score. We would also be happy to provide further responses to any additional questions if needed.
>
> * *The premise of this paper makes little sense. See attached questions. Unless it is revised to meet a basic scientific standard (precise definitions and sensible justifications behind the modeling choices) I won't be able to assess this manuscript.*
>
> The presentation adheres to the standard conventions of machine learning papers, and all definitions and justifications are provided with an appropriate level of mathematical rigor. We acknowledge that certain parts of the reasoning may appear overly simplified to pure mathematicians, but we believe this level of exposition is consistent with papers intended for the broader ICLR community. We also note that the other reviewers recognized the soundness and presentation of our work:
>
> **[fG8R]** “Soundness: 3: good… Presentation: 2: fair… Theoretical Basis: Using the law of total expectation gives a clear mathematical explanation…”
>
>
> **[yoJx]** “Soundness: 2: fair… Presentation: 3: good… Overall, I find the clarity and quality aspects to be strengths of this paper… The clarity and accessibility of writing is a strength of this paper. The exposition is simple and clear, and the writing is easy to follow to a general ML audience. Overall it was a pleasant read… The quality of the mathematical expressions/technical portion is sound to the best of my knowledge.”
>
>
> **[cq4n]** “Soundness: 3: good… Presentation: 4: excellent… Well-written with a fair experimental setup. The paper is clear… Elegant formulation via the law of total expectation.”
>
> * *typo: In line 209 the term P(Z | X in M_test) does not parse. Z is a random variable.*
>
> We apologize for this obvious typo. The correct term should be written as $P(X \in M_\text{test})$. We made a correction in the revised version.
>
> * *Line 97: "we have access to two distinct collections of real data: a train dataset and a test dataset, with the former dedicated for training and the latter for evaluation purposes exclusively. These data can be considered independently drawn from a random variable X acting on a given multidimensional Euclidean data space."
> Line 190: "The following assumptions underpin our approach: (A1) the samples of training data which are selected for evaluation, are contained in two non-trivial disjoint parts1 Mtrain and Mtest of the manifold of data M, respectively."
> These two assumptions are inconsistent. If the training and test datasets are sampled from the same dataset, how can they be contained in disjoint parts of whatever manifold you are talking about? What is the DEFINITION of Mtrain and Mtest?*
>
> In fact, the reviewer’s concern was addressed in the footnote on page 4. We do not assume that $M_{\text{train}}$ and $M_{\text{test}}$ are predetermined before sampling. Instead, for the evaluation samples $x_\text{train}$ and $x_{\text{test}}$, we posit the existence of two non-trivial, disjoint subsets $M_{\text{train}}$ and $M_{\text{test}}$ of the data manifold $M_{\text{data}}$ such that $M_{\text{data}} = M_{\text{train}} \cup M_{\text{test}}$, with $x_{\text{train}} \subset M_{\text{train}}$ and $x_{\text{test}} \subset M_{\text{test}}$. We emphasize that these parts do not require any special structure (specifically, they do not need to be submanifolds or connected sets), are induced solely by the data samples selected for evaluation, and are not predetermined or fixed subsets. Such a formulation enables the application of the simplified form of the law of total expectation as stated in Theorem 1. To ensure clarity and address any potential misunderstandings, assumption (A1) has been rephrased in the revised manuscript.

---

> > ### Author Response · Authors · 2025-11-21
> > **Official Comment by Authors (continued)**
> >
> > * *Line 198: "note that any such Z must implicitly depend on the random variables X and Y —see Equation (5) for an example." No it must not. The random variable Z that is identically equal to zero satisfies E[Z] = 0 but does not "implicitly depend" on any X and Y. There are myriad examples of random variables that have mean zero but do not "implicitly depend" on your X and Y.*
> >
> > The reviewer's concern appears to stem from a misreading of the formulation of assumption (A3). The quoted statement is taken out of context. To recall the full context:
> >
> > “(A3) a baseline metric $M_\text{base}$ capable of capturing the fidelity and diversity of generated samples is defined by the conditional expectation operator, i.e., $M_\text{base} = E(Z|X \in  M_\text{test})$ for some random variable $Z$ with $E(Z) = 0$ iff $p_X = p_G$ (note that any such $Z$ must implicitly depend on the random variables $X$ and $Y$—see Equation (5) for an example).”
> >
> > So, in fact, we assume that the random variable $Z$ satisfies the property $E[Z] = 0$ **if and only if** (abbreviated "iff") $p_X = p_G$. (We recall that $p_X$ and $p_G$ denote distributions of $X$ and $Y$, respectively, see paragraph “Generative Models” in Section 2.) Therefore, the dependence of $Z$ on both $X$ and $Y$ is not merely a consequence of the sole condition $E(Z) = 0$.
> >
> > Moreover, as $Z$ defines (through its conditional expected value) a valid evaluation metric $M_\text{base}$, it also necessitates the dependence of $Z$ on $X$ and $Y$, the random variables representing real and generated data, respectively. If this dependence did not hold (for instance, if $Z=0$) then such a metric would be meaningless. Thus, the concern overlooks this fundamental requirement and the inherent link between $Z$ and the data variables, reflecting a conceptual misunderstanding rather than a substantive flaw in the assumptions.

---

> ### Comment · Reviewer_cq4n · 2025-11-21
>
> I would like to flag that the current review does not appear to follow several essential ICLR reviewing guidelines.
>
> In particular:
> 	•	No summary of the paper’s contributions is provided. The “Summary” section only states that the premise “makes little sense,” without neutrally describing what the paper attempts to do, as required by the guidelines.
> 	•	No strengths or weaknesses are listed, even though the reviewer assigns the lowest possible scores. Both sections simply state “I cannot assess,” which is inconsistent with giving definitive 1/1/1 ratings and a strong reject recommendation.
> 	•	No supporting arguments are offered for the extreme scores. The review mentions two technical issues (train/test inconsistency and a statement about mean-zero variables), but does not explain how these justify “poor” soundness, presentation, and contribution across the entire paper.
> 	•	Very limited constructive feedback is given, despite the guideline to “be positive and constructive” and to “provide additional feedback to help improve the paper.”
>
> I would respectfully ask my peer reviewer to revise their review so that it meets the basic structure and expectations outlined in the ICLR guidelines and provides the authors with a fair and useful assessment.
>
> Thank you.

---

> > ### Comment · Reviewer_zDZj · 2025-11-21
> >
> > I did not understand the basic definitions in this paper, hence the low grades on clarity. As the premise did not parse I couldn't give any higher initial grades on soundness and contribution.
> >
> > The other reviews helped. I see my misconception now. (The point I missed is that Y and M_{train} are dependent while Y and M_{test} are not.) I will revise the review.
> >
> > I would have found the presentation much clearer if the assumptions on the probability model were stated explicitly under the "PALATE" heading on line 201. I now understand that some of these assumptions were made implicitly in the paragraph starting on line 102.

---

### Meta-Review · Area_Chair_D2GB · 2026-01-06

**Summary:**

While Reviewer cq4n praised the method's computational efficiency compared to FLD, the consensus leans towards rejection. The primary critique from Reviewers fG8R and yoJx is that the theoretical contribution is incremental, relying on elementary mathematical reframing rather than a significant conceptual advance.
Reviewer zDZj had an initial misunderstanding that has been resolved in the rebuttal. However, they still note that the presentation could be improved. Two other reviewers recommend rejection (2) based on lack of novelty. Reviewer cq4n scores the paper a weak accept (6), but is ok with rejection.

Overall, the submitted manuscript does not meet the required standards for acceptance.

**Reviewer Concerns:**

See Summary above.

**Reviewer Scores:**

Reviewer zDZj (0): admitted a misunderstanding of the premise, rendering the initial rating uninformative. The reviewer still finds the presentation lacking.

Reviewer fG8R (2): unlikely to change significantly; concerns about incremental novelty remain.

Reviewer yoJx (2): unlikely to change significantly; viewed the math as too elementary.

Reviewer cq4n (6): remains a weak accept based on efficiency.

---

### Decision · Program_Chairs · 2026-01-26

Reject